**RESEARCH**

# Identifying significantly impacted pathways: a comprehensive review and assessment

Tuan-Minh Nguyen[1], Adib Shafi[1], Tin Nguyen[2] and Sorin Draghici[1,3*]

## Abstract

**Background:** Many high-throughput experiments compare two phenotypes such as disease vs. healthy, with the goal of understanding the underlying biological phenomena characterizing the given phenotype. Because of the importance of this type of analysis, more than 70 pathway analysis methods have been proposed so far. These can be categorized into two main categories: non-topology-based (non-TB) and topology-based (TB). Although some review papers discuss this topic from different aspects, there is no systematic, large-scale assessment of such methods. Furthermore, the majority of the pathway analysis approaches rely on the assumption of uniformity of $p$ values under the null hypothesis, which is often not true.

**Results:** This article presents the most comprehensive comparative study on pathway analysis methods available to date. We compare the actual performance of 13 widely used pathway analysis methods in over 1085 analyses. These comparisons were performed using 2601 samples from 75 human disease data sets and 121 samples from 11 knockout mouse data sets. In addition, we investigate the extent to which each method is biased under the null hypothesis. Together, these data and results constitute a reliable benchmark against which future pathway analysis methods could and should be tested.

**Conclusion:** Overall, the result shows that no method is perfect. In general, TB methods appear to perform better than non-TB methods. This is somewhat expected since the TB methods take into consideration the structure of the pathway which is meant to describe the underlying phenomena. We also discover that most, if not all, listed approaches are biased and can produce skewed results under the null.

**Keywords:** Pathway analysis, Signaling pathways, Network topology, Metabolic pathways, Statistical significance, Bias

## Introduction

High-throughput technologies currently enable us to measure gene expression levels of tens of thousands of genes in the scope of a single experiment. Many such experiments involve the comparison of two phenotypes, such as disease vs. control, treated vs. not treated, drug A vs. drug B, etc. Various statistical approaches are subsequently used to identify the genes which are differentially expressed (DE) between these phenotypes, such as *t* test [1], *Z*-score [2], and ANOVA [3]. Although such lists of

genes provide valuable information regarding the changes across phenotypes, and play important roles in the downstream analysis, they alone cannot explain the complex mechanisms that are involved in the given condition.

One of the most common techniques used to address this problem is to leverage the knowledge contained in various pathway databases such as Kyoto Encyclopedia of Genes and Genomes (KEGG) [4], Reactome [5], Bio-Carta [6], NCI-PID [7], WikiPathways [8], and PANTHER [9]. Such pathways model various phenomena as networks in which nodes represent related genes or gene products, and edges symbolize interactions among them based on prior knowledge in the literature. Pathway analysis approaches use available pathway databases and the given gene expression data to identify the pathways which are

*Correspondence: sorin@wayne.edu
[1] Department of Computer Science, Wayne State University, Detroit, 48202 USA
[3] Department of Obstetrics and Gynecology, Wayne State University, Detroit 48202 USA
Full list of author information is available at the end of the article

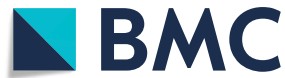

significantly impacted in a given condition. Other complementary approaches include network analysis [10] and GO analysis [11, 12].

Because of the importance of this type of analysis, more than 70 pathway analysis methods have been proposed thus far [11, 13]. These can be divided into two different categories. The first category includes "non-topology-based" methods (non-TB methods, also known as gene set analysis methods), i.e., methods that do not take advantage of the existing knowledge regarding the positions and roles of the genes within the pathways, the directions and types of the signals transmitted from one gene to another, etc.

The first generation in the non-TB category is the *over-representation analysis* (ORA). This approach takes a list of DE genes as input and identifies the pathways in which the DE genes are over- or underrepresented. Some widely used classical approaches from this sub-group use Fisher's exact test [14] and $\chi^2$ test [15]. Many tools that use an ORA approach have been proposed over time, starting as early as 2002: Onto-Express [16, 17], GeneMAPP [18], GeneMerge [19], EASE [20], FuncAssociate [21], etc. Moreover, GO analysis methods, which are classified as ORA, can also be used for pathway analysis. Some popular tools are FatiGO [22], GOstats [23], GOToolBox [24], GoMiner [25, 26], DAVID [27], WebGestalt [28, 29], etc.

The second generation of non-TB approaches includes *functional class scoring methods* (FCS). The hypothesis behind this approach is that small but coordinated changes in sets of functionally related genes may also be important. This approach eliminates the dependency on the gene selection criteria by taking all gene expressions into consideration [30]. Some of the popular FCS approaches are GSEA [31], Catmap [32], GlobalTest [33], sigPathway [1], SAFE [34], GSA [35], Category [36], PADOG [37], PCOT2 [38], FunCluster [39], SAM-GS [40], etc.

Besides ORA and FCS methods, classical statistical tests, such as Kolmogorov-Smirnov test [41] and Wilcoxon rank sum test [42], can also be applied in the context of pathway analysis and fall into the non-TB category.

In principle, considering the pathways as simple unordered and unstructured collection of genes—as the non-TB methods do—discards a substantial amount of knowledge about the biological processes described by these pathways. In essence, all the dependencies and interactions between genes that are meant to capture and describe the biological phenomenon are completely ignored. *Topology-based methods* (TB) have been developed in an attempt to include all this additional knowledge in the analysis. The impact analysis was the first such approach [43]. This was followed by a plethora of over 30 tools and methods that fall in this

category [13] including Pathway-Express [43, 44], SPIA [45], NetGSA [46], TopoGSA [47], TopologyGSA [48], PWEA [49], PathOlogist [50], GGEA [51], cepaORA, cepaGSA [52, 53], PathNet [54], ROntoTools [55], BLMA [56, 57], etc.

Even though there are some review and benchmarking papers which provide some guidance regarding the use of pathway analysis methods, most of these review papers are limited to describing the methods and discussing various characteristics in a theoretical way [13, 58]. Very rarely, some comparisons are done using a few data sets [59], most often simulations. Tarca et al. [60] was arguably the first article that compared 16 different methods using 42 real data sets related to 17 diseases using this type of assessment. However, this comparison is limited to gene set methods (non-TB). A very interesting article by Reimand et al. provided an astonishing perspective on the effect of outdated annotations on pathway enrichment analysis [61] but again comparing the capabilities of the various methods was outside its scope.

Another significant limitation of these review papers attempting to benchmark pathway analysis methods is that they do not take into account the performance of these methods under the null hypothesis, which is the main cause of type I and type II errors in pathway analysis results. Although existing pathway analysis methods work under the assumption that the *p* values are uniformly distributed under the null hypothesis (i.e., that the distributions of the *p* values generated by the pathway analysis methods are uniform), Nguyen et al. [62, 63] showed that this assumption does not hold true for some widely used pathway analysis methods. As a result, the lists of significant pathways provided by these analysis methods often include pathways that are not significantly impacted (false positives), as well as fail to include pathways that are truly impacted (false negatives). None of the existing review papers discusses this major problem.

Here, for the first time, we present a comparison of the performances of 13 representative pathways analysis methods on 86 real data sets from two species: human and mouse. To our knowledge, this is the highest number of real data sets used in a comparative study on pathway analysis methods. The second assessment investigates the potential bias of each method and pathway.

This article provides precise, objective, and reproducible answers to the following important and currently unanswered questions: (i) is there any difference in performance between non-TB and TB methods?, (ii) is there a method that is consistently better than the others in terms of its ability to identify target pathways, accuracy, sensitivity, specificity, and the area under the receiver operating characteristic curve (AUC)?, (iii) are there any specific pathways that are biased (in the sense of being more likely or less likely to be significant across all methods)?, and (iv)

do specific methods have a bias toward specific pathways (e.g., is pathway *X* likely to be always reported as significant by method *Y*)? This article provides some guidance to help researchers select the right method to deploy in analyzing their data based on any kind of scientific criteria. At the same time, this article will be of interest to any computational biologists or bioinformaticians involved in developing new analysis methods. For such researchers, this article is expected to become the benchmark against which any future analysis method will have to be compared. Finally, because of the bias analysis of all known KEGG pathways included here, this article is also expected to be extremely useful to many people involved in the curation and creation of pathway databases.

## Methods

The following subsections will describe briefly the 13 methods studied (Table 1). Eight of these are non-TB methods: Fisher's exact test [14], WebGestalt [28], GOstats [23], Kolmogorov-Smirnov test [41], Wilcoxon rank sum test [42], GSA [35], PADOG [37], and GSEA [31]. The other five of them are TB methods: SPIA [64], ROntoTools [65], CePaGSA, CePaORA [52, 53], and Path-Net [54].

### Non-TB pathway analysis methods

*Fisher's exact (FE) test* is a statistical test that can be used to determine whether two classes of results have a non-random association [14]. In the context of pathway analysis, FE test calculates the probability that an association between the list of DE genes and the genes belonging to a given pathway occurs just by chance. The input of this test, a $2 \times 2$ *confusion matrix*, includes the following four numbers: (i) DE genes belonging to the pathway, (ii) DE genes not belonging to the pathway, (iii) non-DE genes belonging to the pathways, and (iv) non-DE genes not belonging to the pathway. In R, FE test can be performed by using `fisher.test` function.

*WebGestalt* is composed of four modules that allow users to manage the gene sets, retrieve the information for up to 20 attributes for all genes, visualize/organize gene sets in figures or tables, and identify impacted gene sets using two statistical tests, namely the hypergeometric test and Fisher's exact test [28, 29].

*GOstats* uses the hypergeometric probability to assess whether the number of DE genes associated with the term (e.g., GO terms or KEGG pathways) is significantly larger than expected. Similar to other non-TB methods, this computation ignores the structure of the terms and treats each term as independent from all other terms [23].

*Kolmogorov-Smirnov (KS) test* compares two empirical distributions to determine whether they differ significantly [42]. Similar to the FE test, it is a non-parametric test that does not make any assumptions about the distributions of the given data sets. In the context of pathway analysis, the two empirical distributions are the scores of the DE genes inside (denoted as DE-hit) and outside (denoted as DE-miss) a pathway. The null hypothesis here is that there is no association between DE genes and the given pathway, and therefore, there is no significant difference between the two empirical distributions of DE-hit and DE-miss. In R, `ks.test` function can be used where the inputs are the list of DE-hit, DE-miss, their fold changes, and the list of pathway's genes. The output is *p* values of the pathways.

**Table 1** Pathway analysis methods investigated in this study

|  | Method | Category | R-function/package version | Pathway database |
|---|---|---|---|---|
| 1 | Fisher's exact test | non-TB | fisher.test | KEGG v.65 |
| 2 | WebGestalt | non-TB | WebGestaltR 0.3.1 | KEGG v.65 |
| 3 | GOstats | non-TB | 2.48.0 | KEGG v.65 |
| 4 | Kolmogorov-Smirnov test | non-TB | ks.test | KEGG v.65 |
| 5 | Wilcoxon rank sum | non-TB | wilcox.test | KEGG v.65 |
| 6 | GSEA | non-TB | 1.0 | KEGG v.65 |
| 7 | GSA | non-TB | 1.03 | KEGG v.65 |
| 8 | PADOG | non-TB | 1.20.0 | KEGG v.65 |
| 9 | SPIA | TB | 2.30.0 | KEGG v.65 |
| 10 | ROntoTools | TB | 2.6.0 | KEGG v.65 |
| 11 | CePaORA | TB | 0.5 | KEGG (version unknown) |
| 12 | CePaGSA | TB | 0.5 | KEGG (version unknown) |
| 13 | PathNet | TB | 1.18.0 | KEGG v.56 |

Versions of KEGG of CePa methods are unknown because they are embedded in the software
*non-TB* non-topology-based method, *TB* topology-based method

*Wilcoxon rank sum* (WRS) is a non-parametric statistical test generally used to determine whether or not there is a significant difference in the medians of two given populations [42]. In the context of pathway analysis, WRS can be used to compare the ranks or $p$ values (derived from a statistical test, such as a *t test*) of the DE genes inside and outside a pathway. WRS is available in R via the function `wilcox.test`, which takes the list of DE genes, their fold changes, and a list of genes of a given pathway as input. WRS is employed differently by some pathway analysis tools such as SAFE [34] and Camera [66].

*GSEA* uses a KS-like statistic test and considers the entire list of genes rather than simply relying on the cut-off to select the list of DE genes [31]. The GSEA method consists three important steps: (i) calculation of the enrichment score (ES) for each gene set (e.g., pathway), (ii) estimation of the statistical significance of the ES, and (iii) adjustment for multiple hypothesis testing. To derive the ES, it traverses down from the top of the sorted gene list. A running-sum statistic is increased upon encountering a gene inside the pathway and decreased upon encountering a gene outside the pathway. ES is the maximum deviation from zero. Subsequently, a null distribution of the ES is created in the second step using an empirical phenotype-based permutation test. The significance of a pathway is assessed relative to this null distribution. In the last step, normalized ES (NES) of each gene set (pathway) is calculated based on the size of the set. False discovery rate corresponding to each NES is also determined in this final step.

*GSA* was proposed as an improvement of GSEA [35]. First, it uses the "maxmean" statistic instead of the weighted sign KS statistic to derive the gene set score. It also creates a different null distribution for the estimation of false discovery rates. To be more specific, it conducts row (genes) randomization in conjunction with the permutation of columns (samples) and scales the maxmean statistic by its mean and standard deviation to obtain the *restandardized version* of the maxmean statistic.

*PADOG* hypothesizes that genes which appear in fewer pathways have a more significant effect than those which appear in many pathways [37]. Hence, the popular genes are downweighted. Furthermore, PADOG computes gene set scores by assigning the mean of absolute values of weighted moderated gene *t*-scores. Similar to other FCS methods, PADOG's input is the expressions of all the genes under study. The PADOG R package is available at [67].

### TB pathway analysis methods

The first method to be able to incorporate the topological structure of the pathways in the analysis of signaling pathways was proposed in [43]. This is widely known as *impact analysis* and often considered to be the state-of-the-art method in TB pathway analysis. Impact analysis methods calculate the impact of a pathway by combining two types of evidence. The first type of evidence captures the over-representation of DE genes in a given pathway. The second type captures several other important biological factors such as the position and magnitude of expression change for all the DE genes, the interactions between genes as described by the pathway, and the type of interactions. In essence, the measured fold changes of all DE genes are propagated as signals following the topology of the pathway in order to calculate a pathway-level perturbation. The first implementation of impact analysis was Pathway-Express (PE) [43]. Currently, the impact analysis and several follow-up improvements [55, 68] are available in two R packages in Bioconductor [69]: *SPIA* [64] and *ROntoTools* [65].

*CePaGSA* and *CePaORA* consider each pathway as a network where each node can contain one or many genes or proteins [52, 53]. CePaORA only takes the expression changes of the DE genes into account whereas CePaGSA considers the entire list of genes. Both methods consider the whole node as DE if one of the genes residing in the node is DE. Node weights are calculated based on different centrality measurements such as in-degree, out-degree, betweenness, in-largest reach, out-largest reach, and equal weight condition. The pathway score is calculated as a summation of the weights of differentially affected nodes in the pathways. Subsequently, the significance of the pathway is measured based on the null distribution of the pathway score, which is constructed by permutation of the DE genes on a pathway. As a result, for each pathway, there are six different $p$ values derived from the six different measurements mentioned above. Since there is no indication from the original authors about which centrality measurement provides the most accurate result, in this manuscript, we choose the lowest $p$ value of a pathway as its final $p$ value.

*PathNet* relies on two types of evidence in the gene level: direct evidence and indirect evidence [54]. Direct evidence of a gene corresponds to the $p$ value obtained from a statistical test such as a $t$ test when comparing two given phenotypes. Indirect evidence of a gene is calculated from the direct evidence of its neighbor genes in a so-called *pooled pathway*. The pooled pathway is constructed by combining all the pathways in a given pathway database. The PathNet version used in this manuscript incorporates 130 KEGG pathways that were embedded in the software. The $p$ values obtained from these two types of evidence are then combined using Fisher's method [70] to derive a combined evidence for each gene. Finally, the pathway-level $p$ value is computed using a hypergeometric test.

# Results

One of the main challenges in assessing pathway analysis methods is that it is difficult to assess the correctness of whatever comes out from the pathway analysis. Many times, papers describing new methods validate them on only two to three data sets followed by a human interpretation of the results. However, this approach has several problems. First, it is biased and not objective. Living organisms are very complex systems, and almost any analysis result will be supported by some references. Without a deep knowledge of the phenomena involved in the given phenotype, it is impossible to judge objectively whether such connections are really meaningful or not. Second, it is not scientifically sound. A scientific approach should formulate some hypotheses in advance, i.e., what a successful outcome of the pathway analysis should look like. Interpreting and justifying the results obtained from an experiment by searching the supporting literature as evidence are not scientifically sound.

Another approach for benchmarking methods is evaluating them based primarily on their performances on simulated data sets. The problem with this approach is that any simulated data set is constructed based on a set of assumptions, few of which apply to the real data. The resulting comparison not only is difficult to reproduce, but also has some inherent bias.

Here, we introduce two completely objective, reproducible, and scientifically sound approaches to benchmark pathway analysis methods. In the first subsection, we evaluate the methods based on their ability to identify the involved phenotypes using human and mouse benchmark data sets. The second subsection assesses their performances under the true null hypothesis, i.e., there is no true phenotype involved.

## Systematic assessment of the methods using benchmark data sets

### Ability to identify the target pathways on human data sets

A better way of validating a pathway analysis method is assessing its ability to identify the target pathway describing the related mechanism of the condition studied. This validation approach works as follows. First, data sets related to conditions that already have an associated KEGG pathway (i.e., target pathway) are collected. For each experiment, a perfect method would be able to identify the target pathway as significantly impacted and rank it on top. The target pathway is chosen in advance without human interpretation. Hence, this validation is completely objective and scientifically sound. We apply each method on each of those data sets and report the ranks and *p* values of target pathways (Fig. 1).

Here, we use 75 human data sets related to 15 different diseases with each disease being represented by five different data sets to evaluate the ability of methods to identify target pathways. Figure 2 shows violin plots for the rankings (top panel) and *p* values (bottom panel) of the 75 target pathways for each of the 13 competing methods.

On a general note, the median rank of target pathways is within the top half for all methods studied, except for KS (Fig. 2a). None of them, however, has a median rank in the top 20. Notably, the TB methods are more consistent in ranking the target pathways. Specifically, the range of the median rank values obtained by the TB methods (from 45 to 52) is much smaller than the median rank values obtained by the non-TB methods (from 29 to 79). Among the non-TB methods, each of the FCS methods (GSEA, GSA, and PADOG) performs better than any other methods.

Regarding the performance of the individual methods, the best ranks of target pathways were obtained by PADOG (median rank = 29), followed by CePaGSA, ROntoTools, and PathNet which have median rank values of 45, 46, and 46, respectively. This result also confirms the claims in Tarca et al. [37] that PADOG is better than GSEA and GSA.

The *p* values of target pathways using the 13 methods is plotted in Fig. 2b. In contrast to median ranks, median *p* values of non-TB methods are comparable to each other while those of TB methods vary considerably. Among all the methods, the median *p* value obtained by CePaGSA is the lowest (median *p* value = 0.001), followed by PADOG (median *p* value = 0.11) and CePaORA (median *p* value = 0.14).

We also perform a higher level comparison between the ranks and *p* values of the target pathways obtained by non-TB and TB methods. As expected, the median rank values of the TB methods are significantly lower (Wilcoxon *p* value = 8.771E−3) than those of the non-TB methods (Fig. 3a). Similarly, the median *p* values obtained by using TB methods are also significantly lower (Wilcoxon *p* value = 4.51E−4) than those of non-TB methods. These results suggest that overall, in this assessment, TB methods are superior to the non-TB methods.

### Ability to identify the pathways containing the cause of the phenotype on mouse data sets

Although the above assessment is better than the human interpretation approach or using simulated data sets, it still has some limitations: it focuses solely on one true positive, the target pathway. We do not know what other pathways are also truly impacted and therefore cannot evaluate other criteria such as the accuracy, specificity, sensitivity, and the AUC of a method. Here, we use knockout data sets that involve using knockout experiments (KO), where the source of the perturbation is known, i.e., the KO gene. We consider pathways containing the KO gene as positives and the others as negatives. After performing the pathway analysis method on this data set, a

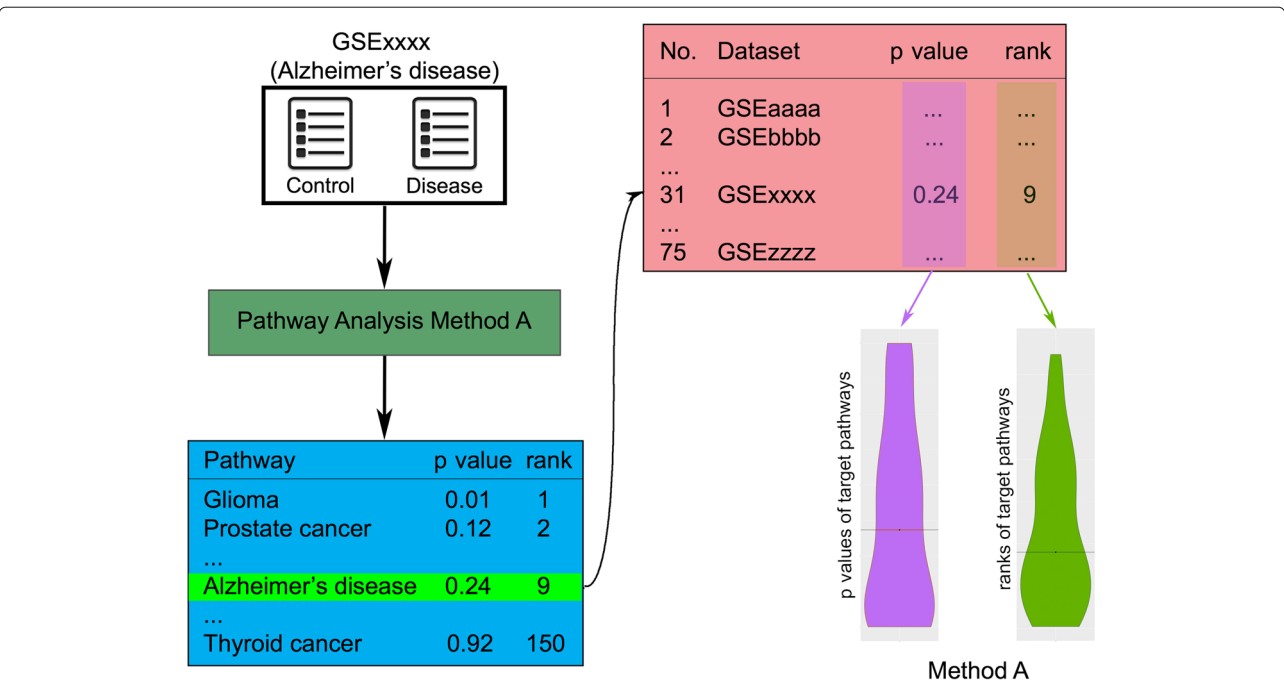

**Fig. 1** The process of evaluating a pathway analysis method based on their ability to identify target pathways. Each pathway analysis method is applied on 75 data sets. Methods are evaluated based on their ability to rank the target pathways. In this example, a data set of Alzheimer's disease is examined, and thus, the target pathway is "Alzheimer's disease." Each method produces lists of ranks and *p* values of the target pathways, which are then used to assess its performance

*p* value threshold of 0.05 is used to determine whether a pathway is significantly impacted. A true positive (TP) is a positive which is correctly identified as significant. Similarly, a true negative (TN) is a negative which is correctly identified as insignificant. A false positive (FP) is a pathway that does not contain the KO gene but is reported as significant. A false negative (FN) is a pathway that contains the KO gene but is not reported as significant.

Subsequently, we calculate the accuracy, sensitivity, specificity, and AUC of methods studied using 11 KO data sets. Since CePaGSA, CePaORA, and Path-Net do not support mouse pathways, they are left out from these comparisons. The comparisons of accuracy, sensitivity, and specificity are illustrated in Additional file 1: Fig. S3. ROntoTools and PADOG have the highest median value of accuracy (0.91). ROnto-Tools also has the highest median value of specificity (0.94). All methods show rather low sensitivity. Among them, KS is the best one with the median value of sensitivity of 0.2.

Among those four statistical measures, the AUC is the most comprehensive and important one because it combines both the sensitivity and specificity across all possible thresholds (Fig. 4). Again, ROntoTools has the highest median value of AUC, namely 0.799, followed by GSEA (0.763) and SPIA (0.719). On the higher level, the AUCs derived by the TB methods are significantly higher

than those derived by the non-TB methods (Wilcoxon *p* value = 0.009).

In conclusion, TB methods outperform non-TB methods in all aspects, namely ranks and *p* values of target pathways, and the AUC. Moreover, the results suggest that there is still room for improvement since the ranks of target pathways are still far from optimal in both groups.

## Investigation of the bias under the null

In this benchmark, we conduct a deeper investigation into the behavior of these methods under the null hypothesis. Here, we create a true null hypothesis by using simulated data sets that are constructed by randomly selected healthy samples from the 75 aforementioned data sets. We apply each method more than 2000 times, each time on different simulated data sets. Each pathway then has an empirical null distribution of *p* values resulting from those 2000 runs (Fig. 5). When the null hypothesis is true, *p* values obtained from any sound statistical test should be uniformly distributed between 0 and 1 [71, 72]. However, *p* values generated from many pathway analysis methods are often unimodal (biased toward 0 or 1) or bimodal (biased toward 0 and 1) (Additional file 1: Figures S4 and S5). More specifically, a null distribution of *p* values of a pathway generated by a method skewed to the right (biased toward 0) shows that this method has a tendency to yield low *p* values and therefore report

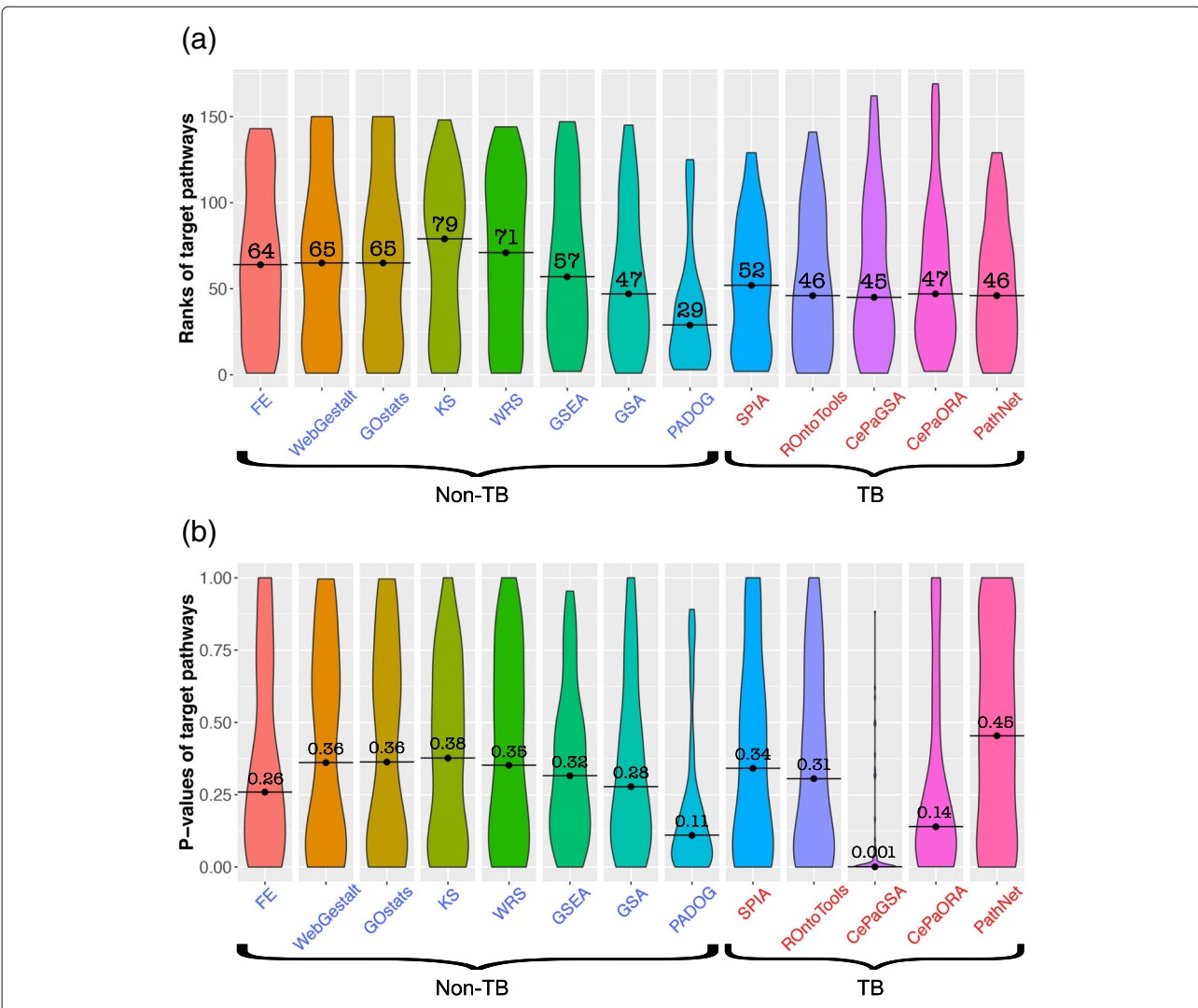

**Fig. 2** The Ranks and *p* values of target pathways derived by 13 methods. We perform each method on 75 human benchmark data sets. The resulting ranks and *p* values of target pathways are plotted in violin plots. The horizontal axis shows the pathway analysis methods in both subfigures. The vertical axis in **a** represents the ranks while the vertical axis in **b** corresponds to *p* values of the target pathways. Hereafter, the labels of non-TB and TB methods are written in blue and red, respectively

the pathway as significantly impacted even when it is not (false positive). By contrast, a null distribution of *p* values of a pathway skewed to the left (biased toward 1) indicates that the given method tends to produce consistently higher *p* values thus possibly report this pathway as insignificant when it is indeed impacted (false negative). The results of this null-hypothesis analysis may explain why some methods work well for certain diseases while they perform poorly for others. If a method is biased to report more often a given cancer pathway as significant, that method may be perceived to perform better in experiments involving that particular type of cancer.

The total number of biased pathways (either toward 0 or 1) produced by these methods are compared in Fig. 6a.

The number of biased pathways is at least 66 for all the methods compared in this work, except GSEA which has no biased pathway. While investigating more, we found that the aggregate *p* values of all the pathways generated by GSEA is uniformly distributed under the null (Additional file 1: Figure S6). A similar conclusion about GSEA was also reached by Nguyen et al. [62].

The number of pathways biased toward 0 produced by 13 methods are shown in Fig. 6b. The figure shows that performing pathway analysis using the FE test produces the highest number (137 out of 150 pathways) of false positives; this is followed by the WRS test (114 out of 150 pathways) and CePaGSA (112 out of 186 pathways). On the other hand, GSEA and PathNet produce no false positive pathways.

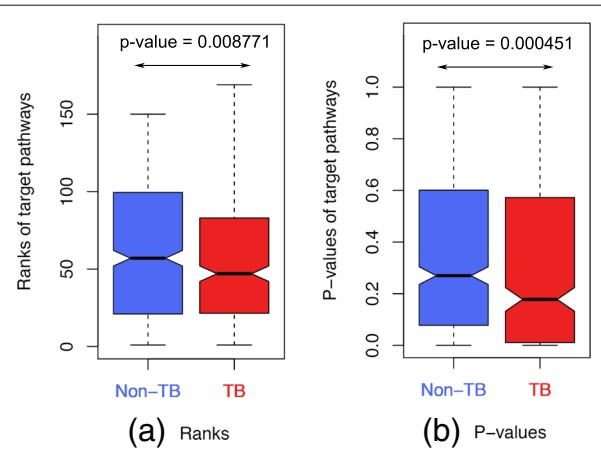

**Fig. 3** The performances of non-TB and TB methods in term of ranks (**a**) and *p* values (**b**) of target pathways. We collect all the ranks and *p* values in Fig. 2 and divide them accordingly into two groups: non-TB and TB methods. Here, lower is better for both ranks and *p* values. The WRS test indicates that TB methods achieved significantly lower ranks (WRS *p* value = 8.771E−3) and *p* values (WRS *p* value = 4.51E−4) than those of non-TB methods

Similarly, the numbers of pathways biased toward 1 produced by different methods are shown in Fig. 6c. PathNet produces the highest number (129 out of 130 pathways) of false negative pathways. No false negative pathways are identified while performing pathway analysis using GSEA, CePaGSA, WRS test, and FE test.

## Discussion

The goal of pathway analysis is to translate the list of genes that are differentially expressed across the given phenotypes (e.g., disease versus healthy, treated versus non-treated, disease subtype A versus disease subtype B, etc.) into meaningful biological phenomena. Over the last few years, more than 70 pathway analysis methods have been proposed. A real problem in the field is the annotation of the pathways. The pathways evolve as more knowledge is gathered. Essentially, at any moment in time, the knowledge captured by the pathways is both incomplete and perhaps partially incorrect. Regardless of the imperfections of today's pathways, one still needs to identify which of these pathways are significantly impacted in the given phenotype. Hence, extensive benchmarking results will be very useful even though the annotations of the pathway will be imperfect at any one particular time. Although there have been already a few publications guiding the users by comparing these methods, they are collectively limited in the following ways: (i) they only discuss the methodological aspects of the methods, (ii) the assessment of the methods is based on simulation data sets which often fail to capture the complexity of real biological phenomena, (iii) they do not compare the performance

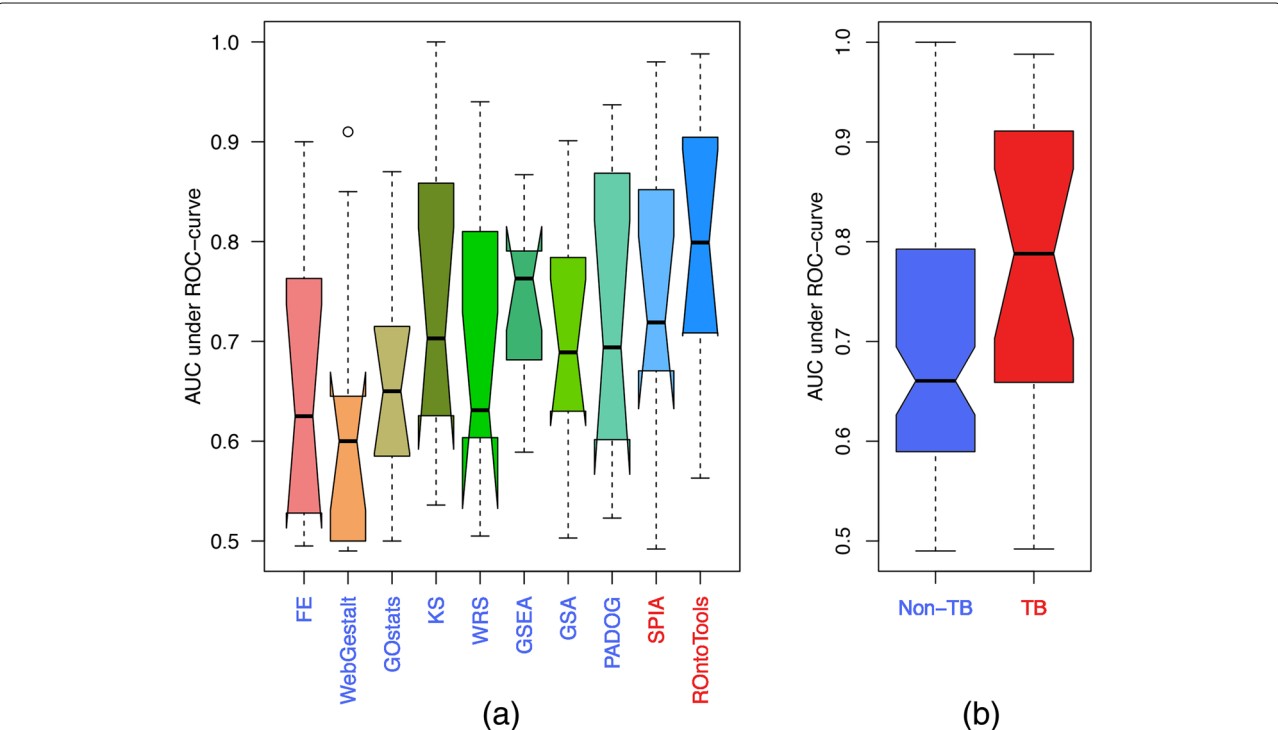

**Fig. 4** The AUCs of eight methods using 11 KO data sets (higher is better). CePaORA, CePaGSA, and PathNet are left out in this comparison because they do not support mouse pathways. ROntoTools has the highest median value of AUC, followed by GSEA and SPIA (**a**). Overall, the AUCs obtained by TB methods are better than those from non-TB ones (Wilcoxon *p* value = 0.009) (**b**)

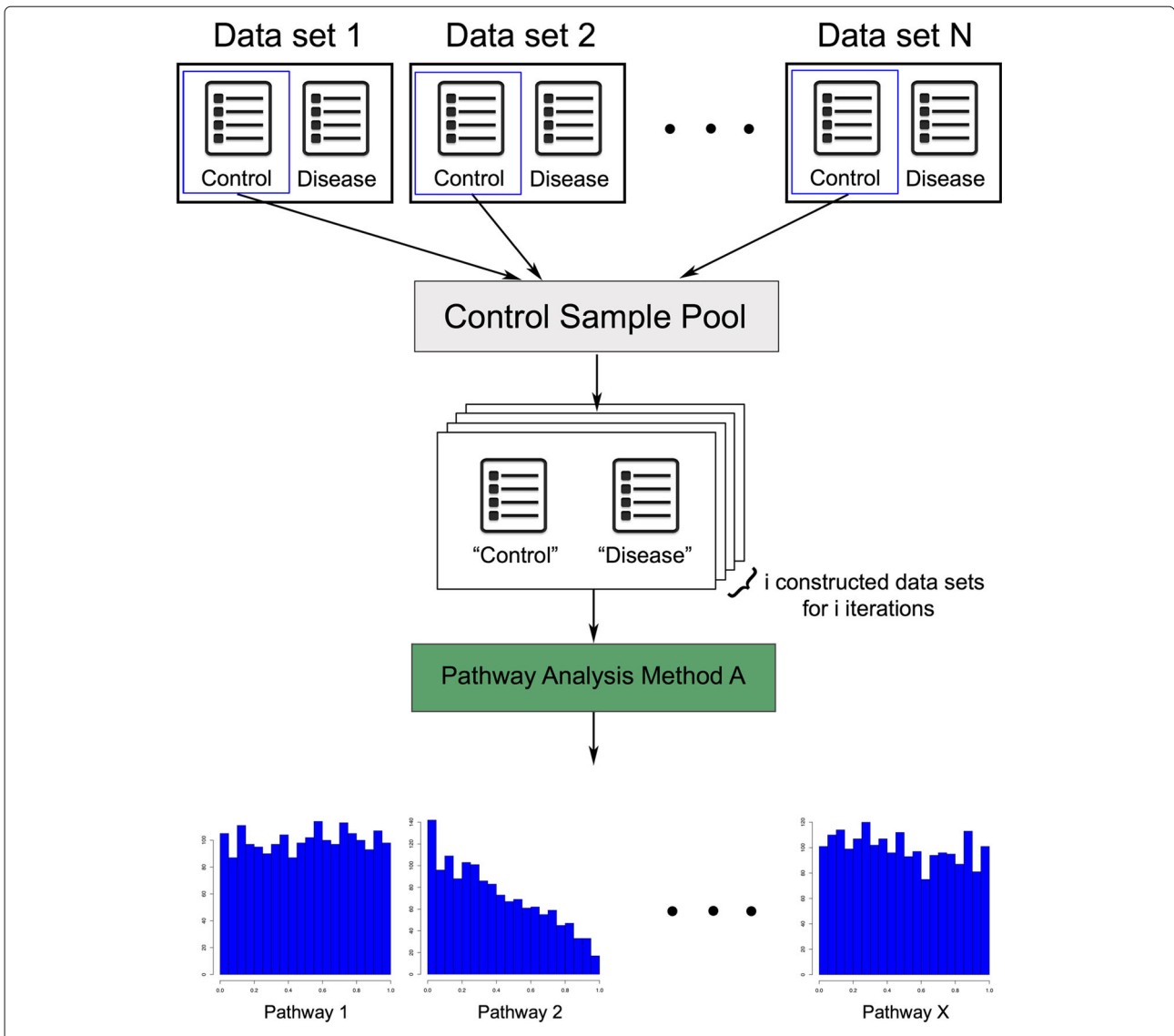

**Fig. 5** The process of creating the null distributions of *p* values for all pathways by a given pathway analysis method. Control samples from data sets are gathered to construct a control sample pool. To create the null distribution of *p* values of all pathways under the null for each method, more than 2000 iterations were performed. The data sets used in these iterations are generated by randomly selecting samples from the control sample pool

of the methods under the null, (iv) they do not take into account the systematic bias of a method introduced by the imbalanced number of data sets for one disease, and (v) they do not take the quality of annotation of the pathways into account, which is one of the real challenge in the field. These limitations may cause significant bias in the conclusions [63]. Here, we address all aforementioned issues and provide a systematic assessment and comparison of 13 widely used pathway analysis methods (8 non-TB and 5 TB methods). Note that all of the R packages of the approaches in this study are non-commercial and free for educational purposes. Therefore, other popular commercial or web service pathway analysis tools (e.g., iPathwayGuide [73], Ingenuity Pathway Analysis [74], or

DAVID [27]) are out of scope of this review. Nevertheless, the results presented here can be extrapolated to these tools as well, based on the approach used. Thus, iPathwayGuide (www.advaitabio.com) uses the impact analysis that is also implemented in ROntoTools so iPathwayGuide results are expected to be comparable with those of ROntoTools. Also, Ingenuity Pathway Analysis and DAVID are both using a hypergeometric test so their results are expected to be comparable with those obtained with Fisher's exact test (FE).

In order to avoid the potential bias in the comparison, we consider several important factors. First, we utilize an equal number of data sets for each disease in our experiment. This is a crucial factor because if a method tends

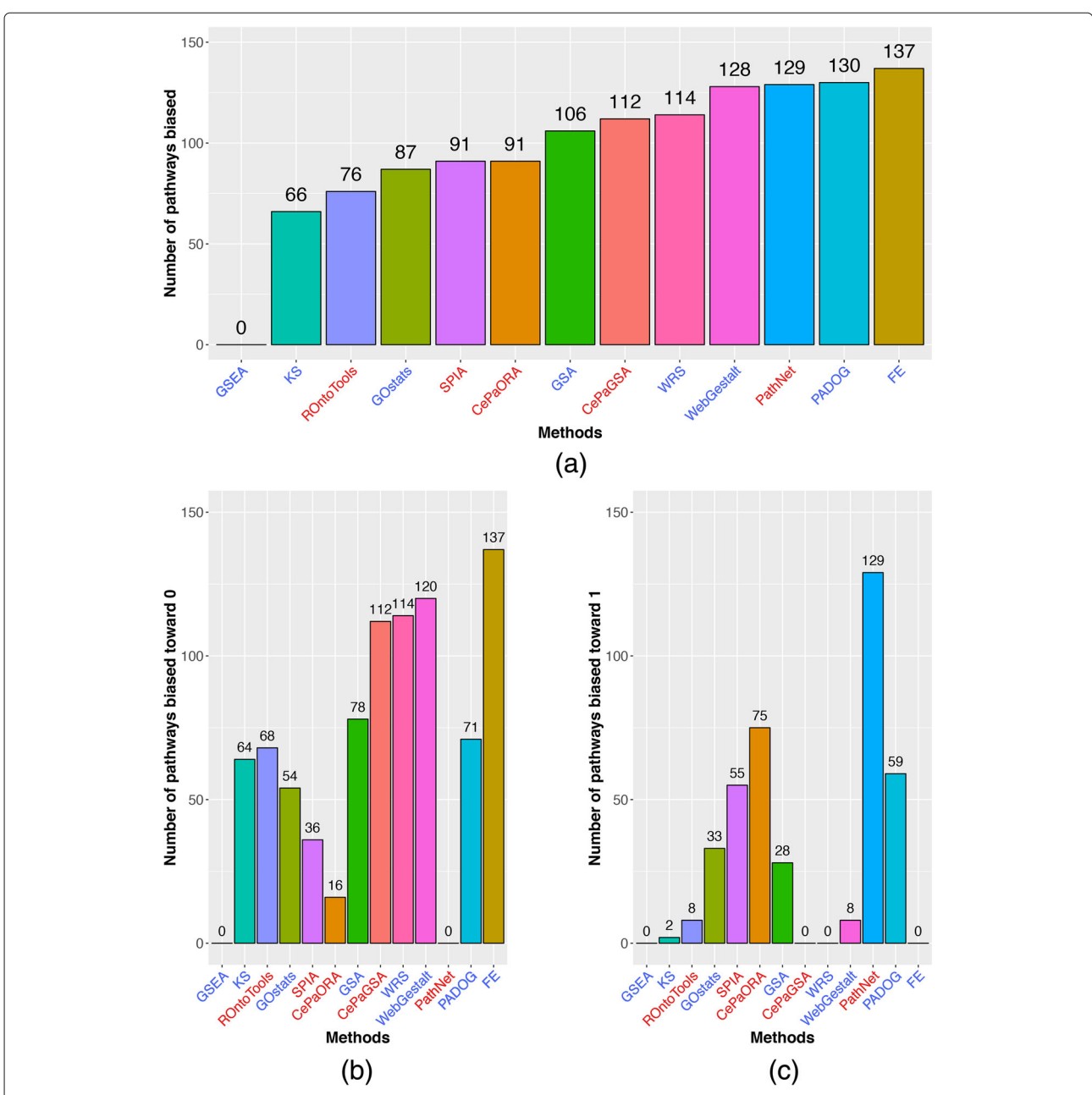

**Fig. 6** The number of biased pathways calculated based on Pearson's moment coefficient. Under the true null hypothesis, an ideal method would produce a uniform distribution of *p* values from 0 to 1 for every pathway. Here, thresholds of Pearson's moment coefficient of 0.1 and − 0.1 are used to determine if the empirical distribution of *p* values is biased toward 0 or 1, respectively. **a** The total number of biased pathways (toward either 0 or 1) produced by each method. Each method, except GSEA, has at least 66 biased pathways. **b** The number of pathways biased toward 0 (false positives) produced by different methods. FE produces the highest number (137 out of 150 pathways) of false positives, followed by WRS (114 out of 150) and CePaGSA (112 out of 186). **c** The number of pathways biased toward 1 (false negatives) produced by different methods. PathNet produces the highest number (129 out of 130) of false negative pathways. The methods in red are TB methods. The methods in blue are non-TB methods

to unsuccessfully identify some pathways associated with some particular diseases as significantly impacted (type II error), then having too many data sets of these diseases will undermine the rank and the performance of this method.

Second, we attempt to reduce the bias caused by different data sets by selecting a fixed number of DE genes, namely 400 DE genes, for each data set (around 10% of total number of genes in KEGG). The classical approach to obtain a list of DE genes from a given gene expression

experiment involves applying thresholds based on *p* values and absolute log-fold changes. However, due to the heterogeneity present in the individual experiments, the number of DE genes obtained from different studies of the same condition often differ significantly [75–77]. For example, with a threshold for the absolute fold change of 1.5 and a threshold for corrected *p* values of 5%, 21 out of 75 human gene expression data sets studied do not have any DE genes. At the same time, one of the data sets has more than 1000 DE genes (Additional file 1: Figure S1). A similar problem occurs with the 11 KO data sets, five of which do not have any DE genes according to these criteria (Additional file 1: Figure S2). This problem in turn makes the downstream analysis (e.g., pathway analysis) inconsistent and biased toward certain data sets. We address this issue by using the same number of DE genes for each data set.

In addition, we apply the use of KO data sets in assessing pathway analysis methods, which has never been used in any comparative study in the field. This approach avoids the shortcoming of the target pathway approach which focuses on the only one true positive, the target pathway. However, a knockout is a severe perturbation of a complex organism, and in some sense, most if not all pathways will be affected to some degree. Given this, the problem becomes philosophical: given that most of all pathways will be affected to some degree, which pathways we want the analysis to identify? Our proposed answer to this is that we want the analysis to identify the pathways that contain the cause of the phenotype, i.e., the KO gene. We feel that this definition is reasonable because it satisfies two conditions: (i) all "interesting" pathways according to the definition above are truly interesting and (ii) there is no other way to define "interesting" pathways without including all other pathways or without using a completely arbitrary decision threshold.

Our assessment using both human and mouse KO data sets shows that the TB methods consistently provide better results than the non-TB methods in terms of ranks and *p* values of target pathways, as well as the AUC.

We also evaluate the performances of pathway analysis methods under the null hypothesis. It is interesting to see that the total number of pathways biased toward 0 is almost double the number of pathways biased toward 1 (696 pathways biased toward 0 versus 356 pathways biased toward 1). In other words, majority of the pathway analysis methods (except GSEA) tend to consider a given pathway as significantly impacted when it is not truly impacted (i.e., to report false positives).

More importantly, benchmarking methods based on their performances under the null overcome the problem of currently poor annotation of the pathways. In other words, when analyzing two groups of healthy samples (the true null hypothesis), a sound method (e.g., GSEA) should not identify any pathway as significantly impacted, regardless of its quality of annotation.

In order to obtain a better understanding of any of these methods, both studies (the systematic assessment of the methods using benchmark data sets, and the investigation of the bias under the null) performed in this manuscript should be considered. A method might perform better than other comparative methods in terms of ranks and *p* values of the target pathways, but that might be due to its intrinsic bias toward 0. For example, PADOG achieves the lowest median rank of the target pathways (Fig. 2a) whereas CepaGSA achieves the lowest median *p* values (Fig. 2b). However, from the second study, it appears that an enormous number of the pathways (71 pathways for PADOG, 78 pathways for CePaGSA) reported by these two methods are biased toward 0 (Fig. 6). In other words, those low *p* values are likely to be associated with false positives most of the time. Similarly, GSEA appears to be extremely unbiased and never yield false positives. However, GSEA also exhibits a low sensitivity, i.e., a reduced ability to identify the true positives.

To choose the best pathway analysis method, one should consider the following four crucial factors in order of importance: (i) *number of biased pathways*; (ii) *ranking of the target pathways*; (iii) *AUC, accuracy, sensitivity, and specificity*; and finally (iv) *p values of the target pathways*. The number of biased pathways is the most important factor since a less biased method would yield fewer false negatives and fewer false positives in the result. The second important factor is the ranking of the target pathways. In contrast to the ranking, an assessment of a method based on the derived *p* values of the target pathways is not as trustworthy because the *p* values are extremely sensitives to these factors. For example, the low median *p* value achieved by CePaGSA is due to the fact that this method reports the majority of the pathways (61.82% in average) as false positives in any given condition.

Choosing appropriate data sets is also a very important but often neglected step while benchmarking pathway analysis methods. The target pathways related to the diseases or conditions of these data sets should have unbiased null distributions of *p* value produced by all methods studied. If the null distribution of *p* values of a target pathway is not available, knowing the probability of that pathway being biased toward 0 or 1 is also helpful. In an attempt to provide this information, for each pathway, we calculate the number of methods (out of the 13 methods investigated) biased toward 0 or 1 (Fig. 7). The resulting graph indicates that there is no such "ideal" unbiased pathway. Each pathway is biased by at least 2 out of 13 investigated methods. Some pathways are biased by as many as 12 methods (out of 13 methods). The common characteristic of these most biased pathways is that they are small in size (less than 50 genes), except for "PPAR signaling

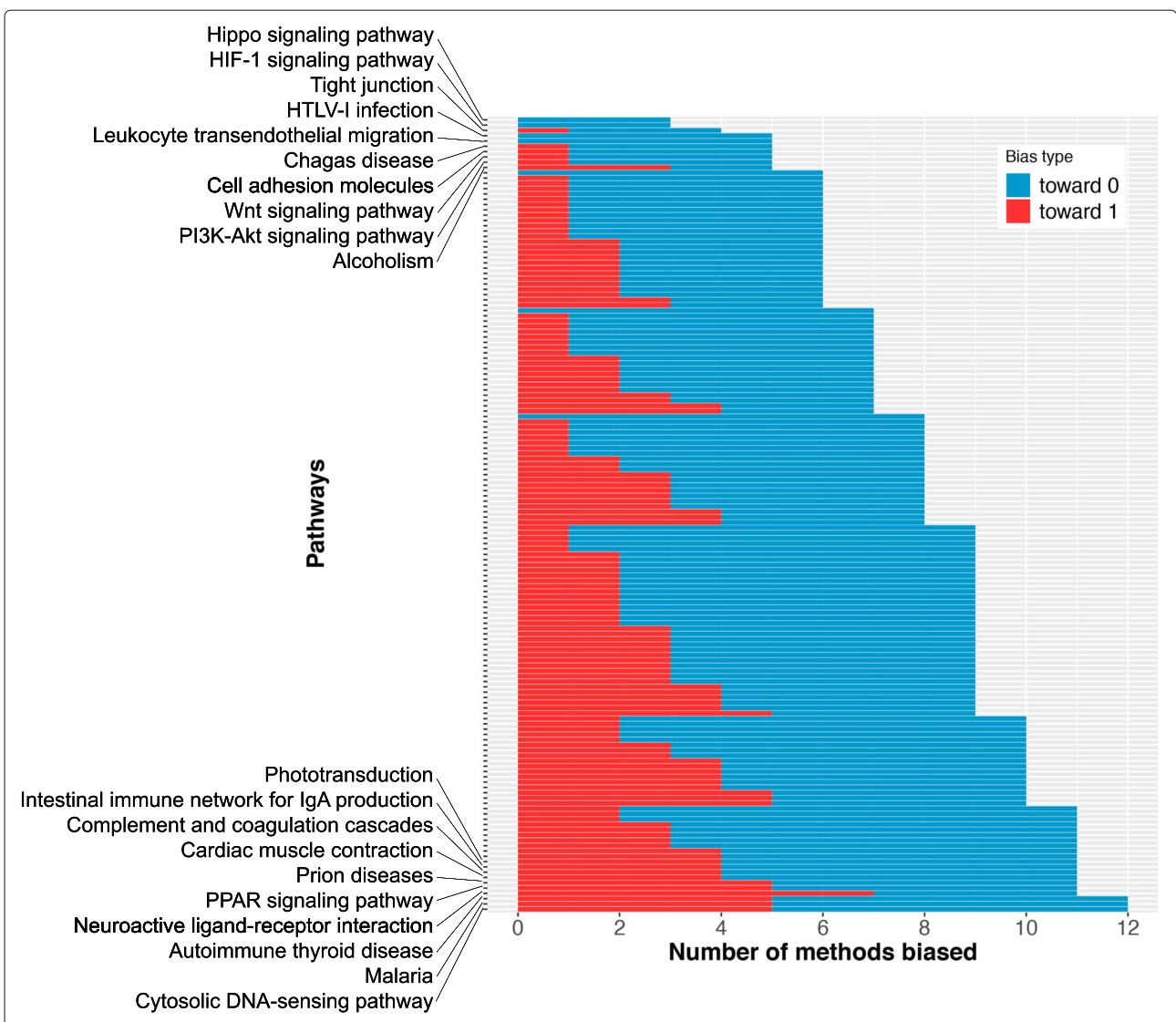

**Fig. 7** The number of methods biased for each pathway. The *y*-axis shows the KEGG pathways, while the *x*-axis indicates the number of methods biased toward 0 and 1, respectively. Each horizontal line represents a pathway. The lengths of the blue and red lines show the number of methods in this study biased toward 0 and 1, respectively. Pathways are sorted by the number of methods biased. There is no pathway that is unbiased for all methods. The top 10 least and top 10 most biased pathways are shown by name

pathway" (259 genes) and "Complement and coagulation cascades" (102 genes). In contrast, all pathways in the top 10 least biased have more than 200 genes and up to 2806 genes. In essence, small pathways are generally more likely to be biased than larger ones. The full list of pathways and their numbers of biased methods is provided in Additional file 1: Table S3.

### Recommendations for pathway analysis users
Based on the extensive testing and comparisons described here, we can provide some guidance for researchers who need to perform a pathway analysis. First and foremost, one should decide what type of analysis they are interested

in. Topology-based (TB) methods provide a better ability to identify pathways that contain genes that caused the phenotype or are closely related to it (such as KO genes, or genes bearing variants that significantly affect their function, etc.). A topology-based analysis is also recommended when (i) it is important to consider how various genes interact, (ii) one wishes to take advantage of the sizes and directions of measured expression changes, (iii) one wishes to account for the type and direction of interactions on a pathway, (iv) one intends to predict or explain downstream or pathway-level effects, and (v) one is interested in understanding the underlying mechanisms. The topology-based approach that provided the

best AUC across our 11 KO data set was the impact analysis, as implemented in ROntoTools [65]. The same impact analysis approach is also used in iPathwayGuide [73, 78].

A non-TB method may be more useful when one needs to analyze arbitrarily defined sets of genes, rather than pathways. In this category, GSEA provided the highest AUC in our extensive testing. GSEA was also the most unbiased method out of the 13 approaches benchmarked in our studies.

The Fisher's exact (FE) test or hypergeometric test is arguably the most widely used method for enrichment analysis. However, our results show that FE is not very suitable in the context of pathway analysis. Figure 6 shows that FE test performs the worst among the 13 compared pathway analysis methods: 137 out of 150 pathways are biased toward 0, that being very likely to often produce false positives. This should be a strong cautionary note to the users of other platforms using this test, such as Ingenuity Pathway Analysis [74] or DAVID [27]. One of the main reasons for the poor performance of the FE test is that it assumes that the genes are independent, while the genes on any pathway influence each other as described by the pathway. Another reason is that the FE test ignores the roles of genes situated in key positions (e.g., a single entry point in a pathway), as well as the number, direction, and type of various signals through which genes on the pathway interact with each other.

## Materials and benchmarking approaches
### Selection of DE genes
In order to select the DE genes, we first calculate the gene-level $p$ values using the two sample $t$ test. Subsequently, we select the genes that have $p$ values less than 5%. Finally, the top 400 genes (around 10% number of genes present in KEGG) with the highest unsigned log-fold changes are considered as DE genes.

### Ranks and *p* values of target pathways
Each data set is associated with a disease or condition whose known mechanisms involved are described in a pathway in KEGG, named *target pathway*. Ideally, a good pathway analysis method would rank the target pathway on top with a small $p$ value. We perform each method on the 75 data sets and put the resulting ranks and $p$ values in the violin plots for the comparison (Fig. 1).

### Statistical measures
In a KO experiment, we consider the pathways containing KO gene as true positives and the other pathways as true negatives. With the definitions of true positives (TP), true negatives (TN), false positives (FP), and false negatives (FN) described in the "Ability to identify the pathways containing the cause of the phenotype on mouse data sets"

section, one can calculate the accuracy, sensitivity, and specificity as follows:

$$\text{Accuracy} = \frac{\text{TP} + \text{TN}}{\text{TP} + \text{FP} + \text{TN} + \text{FN}} \tag{1}$$

$$\text{Sensitivity} = \frac{\text{TP}}{\text{TP} + \text{FN}} \tag{2}$$

$$\text{Specificity} = \frac{\text{TN}}{\text{TN} + \text{FP}} \tag{3}$$

The receiver operating characteristic curve (ROC curve) is a graphical representation of the relationship between the sensitivity and the false positive rate (FPR $= 1 -$ specificity) for every possible $p$ value cutoff, where sensitivity is on the $y$-axis and FPR is on the $x$-axis. The AUC, the area under the ROC curve, is one of the most important evaluation metrics since it measures a test's discriminative ability.

## Performances of methods under the null
### Null hypothesis generation
As a starting point, we combine the control samples from the 75 benchmark data sets to create a *control sample pool*. It is important to stress that this set only contains samples from healthy individuals. For each analysis, we create a simulated data set by randomly choosing 15 samples as "disease" and 15 samples as "control" from the pool. For each of the 13 pathway analysis methods, we create 2000 such simulated data sets and perform pathway analysis separately on each of them, resulting in a list of 2000 $p$ values for each pathway under the null (Fig. 5).

### Metric for bias identification
From all of the non-uniform distributions, we only focus on the ones that are biased toward 0 (right-skewed or positively skewed) or 1 (left-skewed or negatively skewed), since they are responsible for type I and type II errors. We use Pearson's moment coefficient to determine the skewness of a distribution [79]. It is the third standardized moment and is defined as:

$$\gamma_1 = E\left[\left(\frac{X - \mu}{\sigma}\right)^3\right] = \frac{\mu_3}{\sigma^3} \tag{4}$$

where $\mu$ is the mean, $\sigma$ is the standard deviation, $E$ is the expectation operator, and $\mu_3$ is the third central moment.

If $\gamma_1 \simeq 0$, then the distribution of $p$ values is symmetric, i.e., it is unbiased. To decide whether a distribution is biased toward 0 or 1, we set a threshold of $\pm 0.1$. To be more specific, $\gamma_1 > 0.1$ indicates the distribution is right-skewed (biased toward 0) and $\gamma_1 < -0.1$ means it is left-skewed (biased toward 1).

## Additional files

**Additional file 1:** Supplementary figures and tables. **Figure S1.** Distribution of numbers of DE genes of 75 human gene expression data sets in the first experiment using the thresholds of corrected $p$-values < **0.05** and **log**|**FC**| > **1.5**. **Figure S2.** Distribution of numbers of DE genes of 11 mouse gene expression data sets using different thresholds of 1.5 and 5% for **log**|**FC**| and corrected $p$-values, respectively. **Figure S3.** Comparison of 8 methods using 11 KO data sets in term of accuracy, sensitivity, and specificity. **Figure S4.** Examples of pathways that have empirical null distributions of $p$-values biased toward 0. **Figure S5.** Examples of pathways that have empirical null distributions of $p$-values biased toward 1. **Figure S6.** Aggregate $p$-values of all the pathways generated by GSEA are uniformly distributed under the null. **Table S1.** 75 benchmark data sets of 15 diseases used to compare 13 methods in this paper. **Table S2.** Eleven knockout benchmark data sets used to compare 8 methods in this paper. **Table S3.** Number of methods biased for each pathway. (PDF 6475 kb)

**Additional file 2:** Review history. (PDF 116 kb)

### Acknowledgements
We would like to thank Brian Marks and Cristina Mitrea for help and discussions.

### Review history
The review history is available in Additional file 2.

### Authors' contributions
TN, SD, and TMN conceived and designed the project. TMN performed the experiments with the help of TN and AS. TMN, AS, and SD analyzed the data and results. TMN, AS, and SD wrote the paper. All authors read and approved the final manuscript.

### Funding
We acknowledge the financial support from NIH/NIDDK (1R01DK107666-01), Department of Defence (W81XWH-16-1-0516), and National Science Foundation (SBIR 1853207).

### Availability of data and materials
All 75 human data sets (Additional file 1: Table S1) and 11 mouse KO data sets (Additional file 1: Table S2) used in this study are retrieved from Gene Expression Omnibus (http://www.ncbi.nlm.nih.gov/geo).

### Ethics approval and consent to participate
Not applicable.

### Competing interests
The authors declare that they have no competing interests.

### Author details
[1]Department of Computer Science, Wayne State University, Detroit, 48202 USA. [2]Department of Computer Science and Engineering, University of Nevada, Reno, 89557 USA. [3]Department of Obstetrics and Gynecology, Wayne State University, Detroit 48202 USA.

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

## 