## [Review history. (PDF 116 kb) · Genome Biology]

Review History

Comments to author and authors' point-by-point responses to the reviewers' comments:

Reviewer 1

In this manuscript Nguyen et al. assessed the performance of 11 pathway analysis methods with human disease data sets and mouse knock-out data sets. They showed that topology-based methods performed better.

Pros:

- The review of existing methods is comprehensive.
- The use of mouse knock-out data sets is smart.
- Figure 6 is arguably the most interesting results. This helps the interpretation of identified pathways in literature.
- The manuscript is generally well-written and easy to follow.

Response: We appreciate the reviewer for the very positive assessment of this work.

The pathway analysis has been very successful, and a benchmark on it seems not so important. Using multiple methods and only considering the consensus pathways are good enough in most cases.

Response: We agree with the reviewer that pathway analysis has been very successful in the last decade. In fact, more than 70 methods have been developed and more than 15,000 papers related to pathway analysis were published in 2018, according to PubMed. Indeed, as the reviewer suggests, one possibility is to use multiple methods and focus on the pathways found to be significant by all methods used, i.e. the consensus pathways. However, important questions remain such as: how is one supposed to select which of the 70+ available methods to use? Also, the more methods are used, the fewer the pathways that will be in common between all of them.

What is one supposed to do when there are very few or no consensus pathways? Having an extensive set of benchmark results, as the one presented here, can help researchers identify a smaller, more manageable number of methods, from which to seek consensus. Having to analyze the data with 3-4 methods is certainly better than either analyzing the data with tens of methods, or randomly picking 3-4 out of all methods available.

The real problem in the field is the annotation of the pathways.

Response: We agree with the reviewer that annotation is one of the biggest issues of the field. We are fully aware of this issue and extensively discussed in our previous papers:

- Tin Nguyen, Cristina Mitrea, and Sorin Draghici. Network-based approaches for pathway level analysis. Current Protocols in Bioinformatics, 61(1):8 - 25, 2018.

- Yon S Rhee, Valeri Wood, Kara Dolinski, and Sorin Draghici. Use and misuse of the Gene Ontology annotations. Nature Reviews Genetics, 9(7):509 - 515, 2008.

The pathways evolve as more knowledge is gathered. Essentially, at any moment in time, the knowledge captured by the pathways is both incomplete and perhaps partially incorrect. However, this is true about the boundaries of knowledge in most fields of science. Regardless of the imperfections of today ' s pathways, many people still think it is useful to identify those existing pathways that are significantly impacted in a given phenotype and perform a pathway analysis. This is illustrated by the fact that more than 67,000 PubMed papers published in the past 5 years used pathway analysis. We expect that the extensive benchmarking results included here will be very useful to these authors even though the annotation of the pathway will be imperfect at any one particular time.

We mentioned this point in the " Discussion " section to clarify the annotation issue: "A real problem in the field is the annotation of the pathways. The pathways evolve as more knowledge is gathered. Essentially, at any moment in time, the knowledge captured by the pathways is both incomplete and perhaps partially incorrect.

Regardless of the imperfections of today ' s pathways, one still needs to identify which of these pathways are significantly impacted in the given phenotype. Hence, extensive benchmarking results will be very useful even though the annotations of the pathway will be imperfect at any one particular time."

Because of the poor annotation, experimental biologists do not seriously take the results of enrichment analysis into consideration. They will read the papers describing each gene anyway.

Response: We agree that most experimental biologists will read the most important papers related to each of the genes found to be differentially expressed anyway.

However, most biologists will also be interested to know the pathways that are significantly impacted, as well. Very few life scientists will run the risk of publishing their work without ever attempting to perform a pathway analysis when so many tools are readily available. Nobody would like to have their paper delayed because reviewers might ask for such an analysis, or - even worse - a situation in which their experimental results are later re-analyzed by a different group, possibly reporting findings not reported by the original authors. Since some kind of pathway analysis will be done in most cases, it is important to provide information allowing researchers to choose their favorite pathway analysis method(s).

The use of mouse knock-out data sets is nice, but what if the annotation is not accurate? It seems that natural language processing is the computational bottleneck of the field.

Response: As previously agreed, many pathways may be incomplete or contain inaccuracies. The lack of a reliable way of continuously updating the pathways from the literature through natural language processing and/or other means is indeed a severe bottleneck. We also agree with the reviewer that, in principle, some of the annotations may not be accurate. However, it is reasonable to assume that whatever inaccurate information is contained in the existing pathways is randomly distributed and will not going to favor or disadvantage any particular pathway method. Hence, the assessment provided here still provides very useful information regarding the capabilities and bias of the various methods tested.

The left panel of Figure 4 is inconsistent with the right panel? The median of AUC of two TB methods seems 0.75 (from the left panel) instead of 0.8 as shown in the right panel.

Response: We thank you for your very careful reading of the manuscript and great attention to details. The left panel shows the median AUC of SPIA (0.719) and the median AUC of ROntoTools (0.799). On the right panel, the median AUC of TB methods, which consists of SPIA and ROntoTools, is 0.788. Although the median AUC of TB methods is not equal to the mean of two medians (0.759) of its two subsets, it is indeed correct and consistent to the data. Here, the median AUC of two TB methods is actually 0.788 resulting from 11 AUCs of SPIA (0.793, 0.98, 0.978, 0.719, 0.682, 0.63, 0.492, 0.755, 0.659, 0.911, and 0.686) and 11 AUCs of ROntoTools (0.783, 0.988, 0.981, 0.838, 0.799, 0.563, 0.634, 0.898, 0.594, 0.911, and 0.794).

The comparison between TB and non-TB seems to be dependent on the methods selected. While the conclusion is consistent with initiation, I am not sure it is well supported by the data.

Response: We appreciate the comment. The reviewer is correct that the result depends on the methods selected. Since we could not include all the methods available in our study, we chose the ones that are most widely used by the scientific community. For examples, according to Google Scholar GSEA, SPIA, and GSA are cited 16,418 times, 827 times, and 794 times, respectively. In the revised manuscript, we included two other tools from a different domain (GO analysis tools) at the explicit request of one of the reviewers. In the mouse KO tests, which is the most telling in our opinion, we used all methods that support mouse, precisely to avoid introducing any selection bias.

Regarding the data sets, we actually think that it is one of the most crucial aspects in benchmarking pathway analysis methods, or in any comparison study in general. Unfortunately, this step is often neglected, or in some cases, misconducted to obtain a desired result. In our work, we tried our best to suppress any bias that the data sets might cause. First, we increased the number of data sets to 86. This is an order of magnitude higher than the the highest number of real data sets used in any previous comparative study on pathway analysis methods.

The 900 analyses performed on 86 data sets with 2,601 samples are a very significant and unprecedented number of analyses that are expected to be very useful to the readers. Furthermore, precisely in order to avoid any bias, we included the exact same number of data sets for each disease or condition.

The reviewer is correct that the results are mixed since we conducted different experiments and found no method that is superior in all aspects. In term of bias under the null, the performances of both groups are comparable with the Wilcoxon p -value = 0.413. In our view, the data show that TB methods are superior than non-TB methods because of the significant p -values (< 0.05) in the comparisons on both the known disease data sets (yielding both lower ranks and more significant p -values), as well as on the KO datasets (better AUCs). At the end, the paper reports all results obtained for all methods and the readers will be free to interpret the data themselves and possibly reach conclusions that may be different from those drawn in the paper.

The introduction section seems to be a prolonged abstract?

Response: In order to address this issue, we combined the introduction and background sections into a more concise section.

The order of the figures is strange. To put Figure 7 first and then Figure 1 will make Figure 1 much easier to read. Similarly, Figure 8 should be ahead of Figure 5.

Response: We thank the reviewer for the valuable feedback. We rearranged the figures as suggested, and also modified the text accordingly.

It is suggested that the notches be added to the boxplots. This will make the level of statistical significance clearer.

Response: Thank you for your suggestion. The notches are now added to all the boxplots throughout manuscript.

Reviewer 2

The manuscript by Tuan-Minh et al. performed a comprehensive review in the pathway analysis field. Apart from the theoretical method description of each method, they use a bunch of real data across both human and mouse to evaluate the performance of both TB methods and non-TB methods in different statistical aspects while taking a variety of possible and important influence factors into consideration, such as data sets sources and pathway bias of each method under null distribution, which is timely and comprehensive. In general, it does make an important contribution to guide both the bioinformaticians for creating new methods with better performance in the future and analysts to choose better methods for their analysis.

Response: We are so thankful for the positive feedback.

However, there are still several concerns that need to be addressed.

Researchers usually perform pathway analysis and GO analysis simultaneously as also mentioned somewhere in the manuscript to dissect the underlying biological mechanisms given a phenotype. Therefore, it would be more useful if you can also include GO analysis methods into the comparison.

Response: We thank the reviewer for the constructive feedback. It is true that sometimes GO analysis methods are used to perform pathway analysis. In response to the reviewer's suggestion, we added to our manuscript two popular GO analysis methods, namely WebGestalt and GOstats, each with more than 1,000 citations according to Google Scholar, We describe these two methods as follows in the manuscript:

"WebGestalt is composed of four modules that allow users to manage the gene sets, retrieve the information for up to 20 attributes for all genes, visualize/organize gene sets in figures or tables, and identify impacted gene sets using two statistical tests, namely the hypergeometric test and Fisher's Exact test." And "GOstats uses the hypergeometric probability to assess whether the number of DE genes associated with the term (e.g. GO terms or KEGG pathways) is larger than expected. Similar to other non-TB methods, this computation ignores the structure of the terms and treats each term as independent from all other terms."The results of these two methods were also added to all the figures and tables throughout the paper accordingly.

We also mentioned the use of GO analysis methods in pathway analysis and suggested some popular tools in the "Introduction" section as follows: "Moreover, GO analysis methods, which are classified as ORA, can also be used for pathway analysis. Some popular tools are FatiGO, GOstats, GOToolBox, GoMiner, DAVID, WebGestalt, etc."

Using the KO data sets seems a good idea to better assess the performance of each method. However, is it reasonable to consider the pathways without containing the targeted knockout gene as true

negatives? Even if they could be triggered by the true causes, they could still be the significantly affected pathways under the condition.

Response: The reviewer is absolutely correct: in principle, there could be pathways that do not contain the KO gene but are affected by it. However, using the same reasoning a knock-out is a rather severe perturbation of a complex organism and, in some sense, most if not all pathways will be affected to some degree. Given this, the problem becomes philosophical: given that most of all pathway will be affected to some degree, which pathways we want the analysis to identify? Or which pathways are defined as “interesting” ? Our proposed answer to this was: we want the analysis to identify the pathways that contain the cause of the phenotype i.e. the KO gene. We feel that this definition is reasonable because it satisfies two conditions: i) all “interesting” pathways according to the above definition are truly interesting, and ii) there is no other way to define “interesting” pathways without including all other pathways or without using a completely arbitrary decision threshold.

The point raised by the reviewer is very interesting so we also added the following paragraph in the “ Discussion ” section of the revised manuscript:

"In addition, we apply the use of KO data sets in assessing pathway analysis methods, which has never been used in any comparative study in the field. This approach avoids the shortcoming of the target pathway approach which focuses on the only one true positive, the target pathway. However, a knockout is a severe perturbation of a complex organism, and in some sense, most if not all pathways will be affected to some degree. Given this, the problem becomes philosophical: given that most of all pathways will be affected to some degree, which pathways we want the analysis to identify? Our proposed answer to this was that we want the analysis to identify the pathways that contain the cause of the phenotype i.e. the KO gene. We feel that this definition is reasonable because it satisfies two conditions: i) all “interesting” pathways according to the above definition are truly interesting, and ii) there is no other way to define “interesting” pathways without including all other pathways or without using a completely arbitrary decision threshold."

It is better to give a suggestive and clear guidance/choice for the readers as a reference, for instance, in certain circumstances, which method to use and under other conditions, which one should be put priority to.

Response: Thank you for the constructive comment. We added such guidance into the manuscript by recommending one method from each category: "Based on the extensive testing and comparisons described here, we can provide some guidance for researchers in need of pathway analysis. First and foremost, one should decide what type of analysis they are interested in. TB methods provide a better ability to identify pathways that contain genes that caused the phenotype or are closely related to it (such as KO genes, or genes bearing variants that significantly affect their function, etc.). A topology-based analysis is also recommended when: i) it is important to consider how various genes interact; ii) one wishes to take advantage of the sizes and directions of measured expression changes; iii) one wishes to account for the type and direction of interactions on a pathway; iv) when intending to predict or explain downstream or pathway-level effects; and v) when interested in understanding the underlying mechanisms. The topology-based approach that provided the best AUC across our 11 KO data set was the impact analysis, as implemented in ROntoTools.

However, a non-TB method may be more useful when one needs to analyze arbitrarily defined sets of genes. In this category, GSEA provided the highest AUC in our extensive testing. GSEA was also the most un-biased method out of the 13 approaches benchmarked in our studies."